# Robot-Assisted Radical Prostatectomy by Lateral Approach: Technique, Reproducibility and Outcomes

**DOI:** 10.3390/cancers15225442

**Published:** 2023-11-16

**Authors:** Moisés Rodríguez Socarrás, Juan Gómez Rivas, Javier Reinoso Elbers, Fabio Espósito, Luis Llanes Gonzalez, Diego M. Carrion Monsalve, Julio Fernandez Del Alamo, Sonia Ruiz Graña, Jorge Juarez Varela, Daniel Coria, Vanesa Cuadros Rivera, Richard Gastón, Fernando Gómez Sancha

**Affiliations:** Instituto de Cirugía Urológica Avanzada (ICUA), Clínica CEMTRO, 28035 Madrid, Spain; juangomezr@gmail.com (J.G.R.); jre@icua.es (J.R.E.); fabioesposito025@gmail.com (F.E.); lll@icua.es (L.L.G.); dcm@icua.es (D.M.C.M.); jfa@icua.es (J.F.D.A.); srg@icua.es (S.R.G.); daniel.coria@live.cl (D.C.); vcr@icua.es (V.C.R.); fgs@icua.es (F.G.S.)

**Keywords:** robot-assisted radical prostatectomy, radical prostatectomy technique, lateral approach, prostate cancer

## Abstract

**Simple Summary:**

Robotic radical prostatectomy is a treatment for prostate cancer. The lateral approach radical prostatectomy technique allows total preservation of the anterior pubovesical complex, as well as vascular and nervous structures in close contact with the prostate, involved in continence and male potency. We analyzed more than 500 patients undergoing robotic radical prostatectomy using the lateral approach technique, operated by two surgeons at our institution, from January 2015 to March 2021. The technique is reproduced by both surgeons, the oncological and functional results are outstanding with this technique, which means that it is a successful treatment to cure prostate cancer, preserving excellent urinary continence and sexual function.

**Abstract:**

Background: Radical prostatectomy by lateral approach allows performing a prostatectomy through a buttonhole, with direct access to the seminal vesicle and fully sparing the anterior pubovesical complex. Our aim is to show the results of reproducing the technique of robotic radical prostatectomy by lateral approach, in terms of intraoperative, postoperative, oncological and functional parameters. Methods: We analyzed 513 patients submitted to robotic radical prostatectomy by lateral approach from January 2015 to March 2021, operated on by two surgeons in our institution. The oncological and functional results of both surgeons were compared. Results: When comparing both surgeons, the rate of positive surgical margins (PSM) was 32.87% and 37.9% and significant surgical margins (PSM > 2 mm) were 5.88% and 7.58% (*p* = 0.672) for surgeon 1 and surgeon 2, respectively. Immediate continence was 86% and 85% and sexual potency at one year 73% and 72%, with a similar rate of complications for surgeon 1 and 2. Conclusions: Radical prostatectomy by the lateral approach technique with preservation of the anterior pubovesical complex is reproducible and offers good oncological and functional results.

## 1. Introduction

There are different surgical techniques for performing robotic radical prostatectomy for the treatment of localized prostate cancer [1,2,3]. The lateral approach is a technique, developed by Dr. Richard Gaston, is based on the dissection of the prostate through a lateral buttonhole, offering maximum preservation of the anterior pubovesical complex and neurovascular bundles, and keeping the use of thermal energy to a minimum [4,5,6,7]. 

The evolution in imaging techniques such as mpMRI, Positron emission tomography (PET), and microultrasound (MUS), among others, allows better identification of tumours and their localization, increasing the diagnostic yield of biopsy systems and influencing the planning of the surgical strategy [8]. However, surgical technique and surgeon experience impact on positive surgical margins (PSM) and functional and oncologic outcomes [9,10,11,12,13,14]. Some surgeons say that the lateral approach technique is difficult to reproduce, but like any technique [15], reproducing it requires knowledge of the appropriate anatomical landmarks.

The aim of our study is to show the results, in terms of intraoperative, postoperative, oncological and functional parameters, of reproducing the technique of robotic radical prostatectomy by lateral approach. 

## 2. Materials and Methods

We analysed 513 patients submitted to robotic radical prostatectomy by lateral approach from January 2015 to March 2021, operated on by two surgeons in our institution. Surgeon 1, RG (the inventor of the technique and with extensive experience in robotic surgery, 289 patients), and surgeon 2, FGS (224 patients). The primary endpoint is reproducibility of the surgical technique. Intraoperative and postoperative results were assessed as surgical time, bleeding, days of hospitalization, days of catheterization, and complications according to Clavien–Dindo classification. Oncological outcomes include surgical margins, biochemical recurrence (BR) and functional parameters in terms of urinary continence and potency rates. Significant positive surgical margins (significant PSM) were defined as >2 mm [16,17]. 

Patients with clinically localized or locally advanced prostate cancer undergoing RARP with more than one year of follow-up were included, staging was based on bone scan and preoperative CT, if necessary, based on the current European Association of Urology guidelines at the time of surgery. Multiparametric prostate resonance imaging (mpMRI) was requested for all patients unless contraindicated (pacemaker, phobia, contrast allergy). Preoperatively, all patients were assessed using the Briganti’s and Memorial Sloan–Kettering PCa nomograms to measure the risk of lymph node involvement.

Patients without follow-up, no definitive pathology data or missing records were excluded. Patients in whom it was not possible to perform the lateral approach technique due to tumour, prostate, patient characteristics, and surgeon’s choice (Sugeon 1: *n* = 16 and surgeon 2: *n* = 23) were excluded from the final analysis. Patients who required adjuvant or salvage radiotherapy were excluded from urine continence analysis, and patients with erectile dysfunction present before surgery that limited intercourse without response to medication were also excluded from potency analysis.

A comparison of the results of surgeon 1 and surgeon 2 was performed using the Student’s *t*-test for quantitative variables, chi-square for qualitative variables, the Kaplan–Meier and log-rank test for power analysis, continence and biochemical recurrence; SPSS software version 25.0 (IBM Corp., Armonk, NY, USA) was used for statistical analysis.

### 2.1. Surgical Technique

All patients were operated on under general anaesthesia in a modified Trendelemburg position at 30° tilt. Following this technique, the first trocar is placed by direct supraumbilical approach pulling the abdominal wall, unless there is a history of previous abdominal surgery and scars with suspected adhesion. In these cases, the left paramedial trocar must be placed first and the other trocars are placed under direct vision. Once all the trocars are placed, the robot could be mounted. We used the Da Vinci Xi robotic system (Intuitive Surgical Inc, Sunnyvale, CA, USA) in all procedures, while the AirSeal^®^ Intelligent Flow System (CONMED Corporation, Largo, FL, USA) was used for the pneumoperitoneum.

In the classic right lateral approach (Figure 1), the Retzius space is partially preserved by the section of the right umbilical ligament and the peritoneum, keeping the left portion of the bladder in place. Exposition of the right pubovesical ligament and endopelvic fascia was performed with dissection close to the bladder and the prostate wall, keeping the fat tissue in place.

One of the fundamental characteristics of this technique is the direct access to the seminal vesicle, performing the dissection through the triangular space covered by fat and through which the veins of the vesico-prostatic venous plexus run, in a triangular space delimited on its internal side by the bladder, anteriorly by the base of the prostate and laterally by the neurovascular bundle. The venous vessels found in this space are clipped with 3 mm mini-clips. The neurovascular bundle must be dissected carefully and bluntly, avoiding the use of thermal energy. Then, the dissection of the vesicles and ligation of the vas deferens and posterior plane of the prostate could be performed, reaching the bladder neck and allowing perfect preservation of the bladder neck. The anterior apron, composed of fibrous tissue and muscle fibres of the detrusor muscle that runs along the anterior aspect of the prostate, has to be dissected until it reaches the bladder neck cranially and the apex distally, saving and completely preserving the anterior pubovesical complex and the Santorini plexus. In the next step, the entire posterior plane has to be dissected from right to left in the right approach, until the left bandelette is reached and separated. Then, it is only necessary to clip perforating prostatic nourishing vessels at the base and medial–lateral zone. 

The next step is mobilizing the prostate and leaving the pubovesical complex anteriorly. Now, the apex is dissected, with skill and patience, the urethra is sectioned, preserving the sphincter and saving the verumontanum. Urethrovesical end-to-end anastomosis was performed using of V-lock^®^ suture, the first stitch starts at 5 o’clock and continues through the posterior late in a running fashion finishing at 3 o’clock. 

After confirmation of hemostasis and bagging of the specimen, the pneumoperitoneum is deflated and the specimen is removed by supraumbilical incision, usually a 20 Ch Foley catheter will be left without leaving drainage in most cases.

### 2.2. Postoperative Management and Follow-Up

The urinary catheter was removed between 7 and 10 days after surgery and the complications were classified according to the Clavien–Dindo classification. 

Surgical margins, biochemical recurrence (BCR), continence and potency were then evaluated. BCR was understood as two consecutive values of PSA >0.2 ng/mL; urinary continence was assessed on the day of catheter removal (immediate), at 3 (early) and 12 months after surgery. Patients are considered fully continent when they used no pads. Patients have been considered potent after surgery if they could achieve sexual intercourse. Patients taking PDE5 inhibitors to achieve intercourse were considered potent with drugs. Patients for whom intercourse depended on a vacuum erection device, penile injection of alprostadil, or a penile prosthesis were not considered as potent. The rate of erectile function recovery was defined considering only patients who were potent before RARP. 

## 3. Results

Table 1 summarizes the baseline characteristics of the 513 patients (Surgeon 1, *n* = 289, Surgeon 2, *n* = 224). Mean age (years) was 62.12 ± 7.49 (IQR = 56–68) and 63.23 ± 6.51 (IQR = 58–64), PSA (ng/mL) = 7.98 ± 5.89 (5.01–8.21) and 7.41 ± 3.47 (5.00–8.21), prostatic volume (mL) 42.30 ± 21.66 (28–50.5) and 47.27 ± 27.7 (28–50), furthermore 199 (69%) and 159 (71%) were clinically significant prostate cancer (csPCa) (GG 2 or higher) for surgeon 1 and surgeon 2, respectively.

Regarding operative and postoperative results: surgical time (min) was 126.28 ± 36.652 and 180.56 ± 57.427 (*p* < 0.001); the lateral approach was possible in 273 (94.5%) and 201 (90.5%) (*p* = 0.6) patients; intraoperative bleeding (mL) 266.08 ± 125.16 and 372.45 ± 135 mL (*p* < 0.001); lymphadenectomy was performed in 117 (42.2%) and 73 (43.2%); the extracapsular extension was found in 62 (21.4%) and 61 (27.3%); positive lymph nodes 13 (4.5%) and 11 (4.9%). The rate of PSM was 95 (32.9%) and 85 (37.9%) for surgeon 1 and surgeon 2, respectively, and significant PSM (>2 mm) was 17 (5.9%) and 17 (7.6%) (*p* = 0. 6). BCR rate was 34 (11.7%) and 27 (12%) for surgeon 1 and surgeon 2, respectively, the average follow-up period was 1728 ± 89.31 days, while the Clavien–Dindo complication rate >2, was 11 (3.8%) and 5 (2.2%). 

Regarding continence and potency rates, 86% and 85% of patients were fully continent at day 0 of bladder catheter removal (totally dry), 93 and 91% at 1 month and 96 and 98% at 1 year for surgeons 1 and 2, respectively (Figure 2A). Sexual potency rates were 60% and 66% at 3 months, 73% and 72% at 1 year for surgeons 1 and 2, respectively. Kaplan–Meier curves and log-rank test are shown in Figure 2, no difference between the surgeons was found for urinary incontinence *p* = 0.080, erectile dysfunction *p* = 0.2 and biochemical recurrence (BR) *p* = 0.7 (Figure 2C–E). Stage pT3a and pT3b data are shown in Appendix A.

## 4. Discussion

Robotic radical prostatectomy by the lateral approach technique can be reproduced with similar results in oncological and functional terms according to our data. When comparing both surgeons, the rate of positive surgical margins was 95 (32.9%) and 85 (37.9%) and significant PSM was 17 (5.9%) and 17 (7.6%) (*p* = 0. 2) for surgeon 1 and surgeon 2, respectively; immediate continence was 86% and 85% and sexual potency at one year 73% and 72%, with a similar rate of complications for surgeon 1 and 2, respectively. 

Clearly, preservation of the bladder neck and neurovascular bundles plays a direct role in the functional outcomes of continence and sexual potency [18]. However, modern prostatic anatomy studies have shown that most of the nerves involved in penile innervation run anterior to the prostate [19]. 

The technique of radical prostatectomy by lateral approach developed by Dr R. Gaston is characterized by some distinctive steps including the direct approach to the seminal vesicle (classically right), which allows the dissection of the homolateral neurovascular bundle, excellent preservation of the vesical neck of the contralateral neurovascular bundle and the complete preservation of the anterior pubovesical complex [1,4,5,6]. 

Other techniques with preservation of the pubovesical complex and Retzius space have also shown excellent results, indicating that preservation of the pubovesical complex seems to play a crucial role in maintaining sexual potency [14,15,20,21,22,23,24]. However, to our knowledge, this is the first series of patients with reproducible results of the lateral approach technique. 

Asimakopoulos et al. previously reported in a small sample of 30 patients undergoing robotic radical prostatectomy with lateral approach technique 10% PSM, 80% of patients were dry at catheter removal, and after 3 months 73% presented an International Index of Erectile Function score > 17 [4].

De Carvalho et al. published a pubovesical complex preservation technique with a median skin-to-skin operative time of 78 min, BCR of 7% and overall PSM rate of 13.3% and 27% in patients with stage pT3. Immediate continence was 85.9% and 98.4% at one year, potency at one month was 53% and 86% at one year [21].

The Bocciardi Retzius-sparing technique claims PSMs of 10.1%, immediate continence was achieved in 92% of the patients, the 1-year continence rate was 96% and potency was 77% [20]. 

However, according to some authors, these results are extraordinary because, according to data from specialized tertiary care centres, around half of the patients reported altered erectile function before radical prostatectomy, 80% of the patients are totally continent after Foley catheter removal and only 53% recover full sexual function [25]. Furthermore, the same authors state that up to 50% of patients show extracapsular extension at the final pathology specimen, 20–35% seminal vesicle invasion, 35–60% of cT3 patients had PSM at the final pathology regardless of nerve-sparing status and >40% PSM in patients with seminal vesicle invasion [25]. 

The direct approach to the right seminal vesicle may seem strange to surgeons unfamiliar with the technique. Still, with the experience we have gained in reproducing the technique, we think that apart from the bladder neck, posterior border of the prostate and neurovascular bundle, the vesicoprostatic veins that run in this triangular space are crucial as an anatomical landmark to follow the path to the seminal vesicle behind these veins that often need to be clipped with mini-clips and avoiding the use of thermal energy.

Based on the principle that the anterior pubovesical complex is fundamental for the preservation of sexual potency, this lateral approach technique through a buttonhole represents a longitudinal incision which is less disruptive than a transverse incision on the fibres of the anterior apron and the pubovesical complex. However, as a result, a contraindication to performing the technique due to the risk of increased margins in this location, according to the recommendation of its creator, are anterior prostate tumours with extraprostatic extension [15]. 

Of course, the evolution in imaging techniques and biopsy systems has improved the characterization and localization of tumours before surgery [26,27]. Therefore, in our practice we request mpMRI for all patients unless contraindicated, and we also perform a prostate-mapping biopsy protocol based on mpMRI fusion biopsy and MUS [9]. 

With surgical expertise and the best possible knowledge of the location of the tumours, it is possible to perform a personalized surgery tailored to the patient. However, several studies indicate that mpMRI is not an accurate indicator of prostatic extracapsular extension, which influences the surgeon’s decision to unnecessarily perform more aggressive surgeries that do not decrease PSM or result in better oncological results with the risk of worse functional outcomes [10,11,28,29,30]. Some nomograms and algorithms, for example, those recently based on MUS, seem to increase the accuracy of predicting the extracapsular location, but more extensive studies are needed on this matter [31].

Surgical technique and experience do matter and seem to have an impact on oncological results in terms of PSM, but surprisingly the impact of surgical experience in BCR is unclear [10,11]. The number of cases needed to reach a plateau in terms of margins is variable, it might be ~200 cases; however, in highly experienced centres with structured learning and mentoring from a very experienced surgeon, proficiency can be achieved early [10,11]. Thus, we think that, with surgical experience, a seasoned surgeon mastering a technique develops the ability to “run away from the tumour” by sorting out sites of extracapsular extension, reducing surgical margins.

As a single-centre retrospective study, our study has several limitations including inherent selection bias, data collection bias, no use of validated questionnaires to objectively measure functional outcomes, no long-term oncological outcomes. Although the results are similar between the two surgeons, we have not performed a stratified analysis of the patients, nor randomisation or propensity score matching techniques, which will be the subject of subsequent studies. 

However, we compare the results of a surgeon who is the creator of the technique, with very extensive experience (more than 5000 cases of robotic surgery), which would be the best possible reference for the procedure, and another surgeon who reproduces the technique learned first-hand directly from surgeon 1. 

## 5. Conclusions

Radical prostatectomy by the lateral approach technique with preservation of the anterior pubovesical complex is reproducible and offers good oncological and functional results.

## Figures and Tables

**Figure 1 cancers-15-05442-f001:**
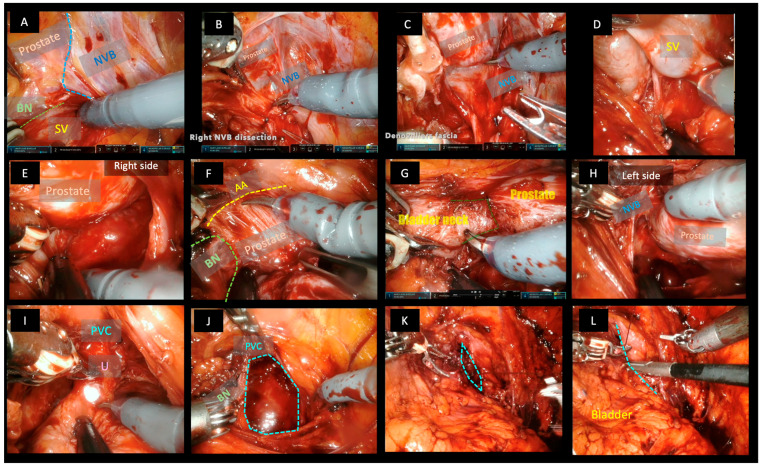
Robotic radical prostatectomy by the lateral approach technique. (**A**) direct access to the seminal vesicle in the triangle formed medially by the bladder neck, anteriorly by the prostate, laterally by the pedicle and nerovascular bundle. (**B**) dissection of neurovascular bundle. (**C**) dissection of right lateral face. (**D**) dissection of seminal vesicles. (**E**) dissection of posterior face from right to left. (**F**) dissection of the plane between the Detrusor’s Apron and the anterior aspect of the prostate. (**G**) dissection and preservation of the bladder neck. (**H**) dissection of the left surface of the prostate and left NVB. (**I**) dissection of the apex preserving the anterior pubovesical complex. (**J**–**L**) closing the buttonhole (BN: bladder neck SV = seminal vesicle NVB = neurovascular bundle AA = anterior apron PVC = anterior pubovesical complex U = urethra).

**Figure 2 cancers-15-05442-f002:**
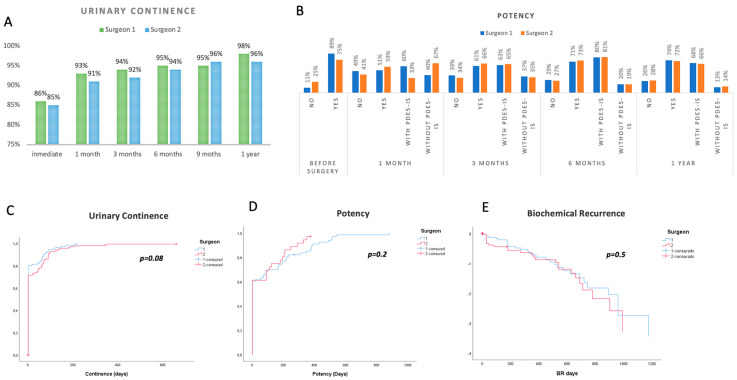
(**A**) Urine continence rates, (**B**) potency. Kaplan–Meier curves and log rank of (**C**) incontinence. (**D**) potency. (**E**) biochemical recurrence, comparing both surgeons.

**Table 1 cancers-15-05442-t001:** Pre- and postoperative results *n* = 513 patients submitted to robot-assisted radical prostatectomy by lateral approach.

	Surgeon 1 (*n* = 289)	Surgeon 2 (*n* = 224)	*p*
Baseline characteristics:			
Age, mean ± SD, IQR	62.12 ± 7.498 (56–68)	63.23 ± 6.512 (58–64)	0.84
PSA ng/mL, ± SD, IQR	7.98 ± 5.89 (5.01–8.21)	7.41 ± 3.47 (5.00–8.21)	0.283
Prostate volume gr, ± SD, IQR	42.30 ± 21.66 (28–50.5)	47.27 ± 27.7 (28–50)	0.077
ED, *n*%	11.10%	25.00%	0.001
csPCa (ISUP ≥ 2), *n*%	199 (69%)	159 (71%)	0.121
TRUS /Fusion Bx/Mapeo (MRI + MicroUS)	117/80/90	38/82/103	<0.001
Postoperative Outcomes:			
Surgical Time min ± SD, IQR	126.28 ± 36.652	180.56 ± 57.427	<0.001
Lateral approach, *n*%	273 (94.5%)	201 (90.5%)	0.650
Lymphadenectomy; Yes, *n*%	117 (42.23%)	73 (43.19%)	0.84
NVB preservation; No/Unilateral/Bilateral, *n*%	25 (8.66%)/76 (26.35%)/188 (64.98%)	32 (14.2%)/29 (13.01%)/163 (72.78%)	<0.001
Intraoperative Bleeding ml ± SD, IQR	266.08 ± 125.16	372.45 ± 135 mL	<0.001
Conversion to open/laparoscopy, Yes/No, *n*%	0	1	-
Hospital stay days, mean ± SD(IQR)	2.84 ± 0.744 (2–3)	3.34 ± 1.022 (3–4)	0.90
Blood Transfusion, Yes/No, *n*%	9 (3.11%)	5 (2.23%)	0.170
csPCa (ISUP ≥ 2), *n*%	271 (93.1%)	207 (92.4%)	0.681
Extracapsular extension, *n*%	62 (21.4%)	61 (27.23%)	0.273
Positive lymp nodes, *n*%	13 (4.49%)	12 (4.9%)	0.533
PSM focal/significant, *n*%.	95 (32.87%)/17 (5.88%)	85 (37.9%)/17 (7.58%)	0.262
BR: Persistence/BR	9 (3.11%)/34 (11.7%)	8 (3.57%)/27 (12.05%)	0.815
Complications (Clavien III/IV), *n*%.	11 (3.8%)	5 (2.23%)	0.763

## Data Availability

The data presented in this study can be shared up on request.

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
