# Peer review of "Robot-Assisted Radical Prostatectomy by Lateral Approach: Technique, Reproducibility and Outcomes"

_cancers, 2023, doi:10.3390/cancers15225442_

Round 1

Reviewer 1 Report

Comments and Suggestions for Authors

cancers-2627367

The authors present a retrospective study describing a new technique for RARP.

 The structure is correct but some questions were raised while studying it.

Materials and methods:

2.1 Surgical technique

71-118. Sometimes the authors write “is” and other times they write “was”. Please use the same tense.

108. “and the complications were” sounds better.

Finally, it is a well prepared study. Unfortunately, I could not understand the endpoints. Were they the advantages of the technique or the learning curve? Furthermore, it would be better if it was a video presentation.

Also, I didn’t find a table with abbreviations attached.

Comments on the Quality of English Language

cancers-2627367

The authors present a retrospective study describing a new technique for RARP.

 The structure is correct but some questions were raised while studying it.

Materials and methods:

2.1 Surgical technique

71-118. Sometimes the authors write “is” and other times they write “was”. Please use the same tense.

108. “and the complications were” sounds better.

Finally, it is a well prepared study. Unfortunately, I could not understand the endpoints. Were they the advantages of the technique or the learning curve? Furthermore, it would be better if it was a video presentation.

Also, I didn’t find a table with abbreviations attached.

Author Response

Reviewer 1 Reply

Thanks to the reviewer for taking the time to review the manuscript, we greatly appreciate the corrections and suggestions.

Comments and Suggestions for Authors

cancers-2627367

The authors present a retrospective study describing a new technique for RARP.

 The structure is correct but some questions were raised while studying it.

Materials and methods:

2.1 Surgical technique

71-118. Sometimes the authors write “is” and other times they write “was”. Please use the same tense.

R= Thanks for the observation, the tense of the text has been modified, we hope it will be more understandable now.

  1. “and the complications were” sounds better.

R= Thanks for the suggestion, it has been corrected in the manuscript.

Finally, it is a well prepared study. Unfortunately, I could not understand the endpoints. Were they the advantages of the technique or the learning curve? Furthermore, it would be better if it was a video presentation.

R= Thank you very much for your comment, our apologies for the lack of explanation, the aim and endpoints are to show our results by reproducing the technique, evaluating reproducibility, surgical time, margins, biochemical recurrence as oncological indicators, as well as continence and potency. About the advantages of the technique, subjectively it offers excellent tissue preservation including neurovascular bundles, nerves, bladder neck and especially the anterior pubovesical complex, which theoretically translates into excellent functional results, like other techniques can also offer, such as Retzius sparing. About the learning curve, there are not many previous studies because it is a little reproduced technique, we are very lucky that we learned directly from its creator Dr Richard Gaston, so the learning could be different if it is learned individually. We hope to be able to contribute with more works and useful materials so that other groups can learn it if they are interested. Furthermore, video presentation has not been the focus of this article, we have several videos of the technique and hopefully they will be the subject of further work.

Also, I didn’t find a table with abbreviations attached.

R= Thank you very much for the observation, abbreviations table is not required in this journal.

Reviewer 2 Report

Comments and Suggestions for Authors

The manuscript compared the oncological and functional outcomes between two surgeons at a single institution for RALP by the lateral approach (with one surgeon being the inventor of the technique) and came to the conclusion that the lateral approach is a reproducible technique. The major criticism of the study is both authors are from the same institution so that it is difficult to assess the external validity of the comparison. A more prudent comparison would be to compare the outcome from surgeons across different institutions. The authors cited several reports from other groups using the lateral approach in the discussion, so they should potentially compare the their outcome with that reported by other groups to prove the lateral approach technique is indeed reproducible.

Minor points

1. Table 1, need to correct for values for ED. Also need to change several "," to "." to make it consistent

2. Table 2 Panel B, please elaborate on the meaning of columns "with PDE5i" and "without PDE5i" and how they are related to the "Yes" and "No" columns. 

Comments on the Quality of English Language

Consider proofreading by a native speaker to improve the quality of the manuscript..

Author Response

Thanks to the reviewer for taking the time to review the manuscript, we greatly appreciate the corrections and suggestions.

The manuscript compared the oncological and functional outcomes between two surgeons at a single institution for RALP by the lateral approach (with one surgeon being the inventor of the technique) and came to the conclusion that the lateral approach is a reproducible technique. The major criticism of the study is both authors are from the same institution so that it is difficult to assess the external validity of the comparison. A more prudent comparison would be to compare the outcome from surgeons across different institutions. The authors cited several reports from other groups using the lateral approach in the discussion, so they should potentially compare the their outcome with that reported by other groups to prove the lateral approach technique is indeed reproducible.

Minor points

  1. Table 1, need to correct for values for ED. Also need to change several "," to "." to make it consistent

R= Thank you very much for your comment, we have corrected it in table.

  1. Table 2 Panel B, please elaborate on the meaning of columns "with PDE5i" and "without PDE5i" and how they are related to the "Yes" and "No" columns. 

R= Thank you for your question, in table 2 panel b, we show potent patients, of the potent ones we show the percentage of those who needed medication (PDE5i) to achieve an effective erection/intercourse. We hope this explanation clarifies your question, thank you very much.

Comments on the Quality of English Language

Consider proofreading by a native speaker to improve the quality of the manuscript..

R= Thank you very much for the comment, a native speaker has improved the English quality of the manuscript.

Reviewer 3 Report

Comments and Suggestions for Authors

In the present paper, the authors compare the outcomes of robotic radical prostatectomy by using a lateral approach .

Specifically, they compare the results of the technique’s inventor to those of another surgeon.

Overall, the paper is well written

However, there are some issues that hinder the publication as it is

First, the authors should provide more information about the recruitment methodology:

-        How did they select patients to be operated by surgeon 1 or 2 ?

-        Were they randomised?

-        Were they conecutive patients?

-        Were patients in the learning curve of surgeon 2 included into analysis?

-         

Again, PSM rate looks likes pretty high. may this figure be related to this specific surgical approach? Specifically, what about the rate in pT2 tumours ?

In the results section, please report p values to a single significant figure unless the p value is close to 0.05, in which case, report two significant figures (for example, 0.046, 0.3, etc.).

Again, please report rates and probabilities to two significant figures (for example, 76%, 2.4%, 0.11%, etc.)

Author Response

Infinite thanks to the reviewer for taking the time to review our manuscript, we highly appreciate it. 

In the present paper, the authors compare the outcomes of robotic radical prostatectomy by using a lateral approach.

Specifically, they compare the results of the technique’s inventor to those of another surgeon.

Overall, the paper is well written

However, there are some issues that hinder the publication as it is

First, the authors should provide more information about the recruitment methodology:

-        How did they select patients to be operated by surgeon 1 or 2 ?

-        Were they randomised?

-        Were they conecutive patients?

R= Thank you very much for your kind words and comments. The patients were not randomized, they are consecutive patients, who met the inclusion criteria during the indicated period from 2015 to 2021.

-        Were patients in the learning curve of surgeon 2 included into analysis?

R= Yes, patients in the learning curve of surgeon 2 were included in the analysis. This could influence the results, but also we think it shows that it is possible to reproduce the technique successfully.

Again, PSM rate looks likes pretty high. may this figure be related to this specific surgical approach? Specifically, what about the rate in pT2 tumours ?

R= The rate of PSM was 32.87% and 37.9 % for surgeon 1 and surgeon 2 respectively, which is in line with results from other series that have reported up to 50% PSM. Importantly, the rate of significant PSM (>2mm) 5.88% and 7.58% (p= 0. 6). On the other hand, several studies have focused on PSM and do not necessarily correlate with BR, and interestingly PSM seems to be surgeon dependent despite knowledge of preoperative imaging. In stage pT2 the PSM rate is 30.7% and 33.6%, the significant PSM rate is 4.25% and 3.57% for surgeon 1 and 2 respectively. The rate of margins in pT3 and the corresponding table can be found as supplementary material.

In the results section, please report p values to a single significant figure unless the p value is close to 0.05, in which case, report two significant figures (for example, 0.046, 0.3, etc.).

R= Thank you very much for the suggestion, we have made the changes in the results section and figures, accordingly to the reviewer's suggestion.

Again, please report rates and probabilities to two significant figures (for example, 76%, 2.4%, 0.11%, etc.)

R= Thank you very much, we have made the changes in the results section and the figures, according to the reviewer's suggestions.

Round 2

Reviewer 1 Report

Comments and Suggestions for Authors

Please specify the primary endpoints. Surgical technique, learning curve or both?

Author Response

Thank you for the comment and suggestion. The primary endpoint is reproducibility of the surgical technique and we have added this to the manuscript in Materials and methods line 47-48.